# Synthesis of Silver Nanoparticles Using *Aggregatimonas sangjinii* F202Z8^T^ and Their Biological Characterization

**DOI:** 10.3390/microorganisms11122975

**Published:** 2023-12-13

**Authors:** Yong Min Kwon, Eun-Seo Cho, Kyung Woo Kim, Dawoon Chung, Seung Seob Bae, Woon-Jong Yu, Jaoon Young Hwan Kim, Grace Choi

**Affiliations:** 1Department of Microbial Resource, National Marine Biodiversity Institute of Korea, Seocheon 33662, Chungcheongnam-do, Republic of Korea; jichi9@mabik.re.kr (Y.M.K.); silverstop20@mabik.re.kr (E.-S.C.); dwchung@mabik.re.kr (D.C.); ssbae@mabik.re.kr (S.S.B.); woonjong_yu@mabik.re.kr (W.-J.Y.); jaoonkim@mabik.re.kr (J.Y.H.K.); 2Department of Natural Products, National Marine Biodiversity Institute of Korea, Seocheon 33662, Chungcheongnam-do, Republic of Korea; kimkw79@mabik.re.kr

**Keywords:** silver nanoparticles, *Aggregatimonas sangjinii*, antibacterial activity, intracellular biosynthesis, minimum inhibitory concentration

## Abstract

The aim of this study is to describe the general features and eco-friendly biosynthesis of silver nanoparticles (AgNPs) from the marine bacterium *Aggregatimonas sangjinii* F202Z8^T^. To the best of our knowledge, no previous study has reported the biosynthesis of AgNPs using this strain. The formation of AgNPs using F202Z8^T^ was synthesized intracellularly without the addition of any disturbing factors, such as antibiotics, nutrient stress, or electron donors. The AgNPs were examined using UV–vis spectrophotometry, transmission electron microscopy, energy-dispersive X-ray spectroscopy, nanoparticle tracking analysis, and Fourier transform infrared spectrometry. The UV–vis spectrum showed a peak for the synthesized AgNPs at 465 nm. The AgNPs were spherical, with sizes ranging from 27 to 82 nm, as denoted by TEM and NTA. FTIR showed various biomolecules including proteins and enzymes that may be involved in the synthesis and stabilization of AgNPs. Notably, the AgNPs demonstrated broad-spectrum antibacterial effects against various pathogenic Gram-positive and Gram-negative bacteria, including *Escherichia coli*, *Bacillus subtilis*, and *Staphylococcus aureus*. The minimum inhibitory concentrations and minimum bactericidal concentrations of the F202Z8^T^-formed AgNPs were 80 and 100 µg/mL, 40 and 50 µg/mL, and 30 and 40 µg/mL against *E. coli*, *B. subtilis*, and *S. aureus*, respectively. This study suggests that *A*. *sangjinii* F202Z8^T^ is a candidate for the efficient synthesis of AgNPs and may be suitable for the formulation of new types of bactericidal substances.

## 1. Introduction

Nanoparticles (NPs) are defined as particles with an average size between 1 and 100 nm. Nanoparticles have become increasingly important in various fields, including medicine, cosmetics, agriculture, and food sciences [1], and can be synthesized chemically, physically, and biologically [2]. Physical and chemical methods for NP synthesis have been utilized to enhance the synthesis efficiency since the beginning of the 20th century [3]. However, conventional methods of NP synthesis require the use of hazardous materials, advanced equipment, high heat generation, and high energy consumption, and harm the environment [4,5]. Among these approaches, biological NP synthesis, using microorganisms and plants, has been widely reported and is recognized as an efficient and green method for the utilization of microorganisms as nanofactories [6]. Green methods for NP synthesis have great potential as easy, eco-friendly, cost-effective, and nontoxic tools that do not require the use of high energy, pressure, temperature, or harmful chemicals compared with conventional methods [5,7,8]. Metallic NPs synthesized either intracellularly or extracellularly offer benefits such as easy handling, straightforward downstream processing, rapid scalability, and convenient genetic modification [9].

The use of diverse marine microorganisms for the green synthesis of NPs has received considerable attention because of their ability to adapt to harsh marine environments as well as a broad range of temperatures, salinity, and pH [10,11]. The unique conditions of these habitats often lead to the development of microorganisms with extraordinary capabilities in NP biosynthesis. However, the challenge remains of identifying specific marine strains that offer distinct advantages over terrestrial microorganisms in NP synthesis. Numerous marine microorganisms play crucial roles in the formation of nanostructured mineral crystals and metallic NPs. These biomaterials exhibit remarkable characteristics, including shape, size, arrangement, and composition, which are comparable to those produced using chemical methods. Research has been conducted on the use of various microorganisms, particularly bacteria and fungi, for the synthesis of a wide range of metal NPs, including silver, gold, zinc, titanium, copper, alginate, and magnesium [12]. The bacterial green synthesis of NPs provides advantages, including easy handling, scaling up, and manipulation. Moreover, bacteria contain a lot of bioactive substances, such as enzymes, proteins, pigments, and polysaccharides, that can be used as effective reducing and capping agents in the NP synthesis process [13,14].

Among these NPs, silver NPs (AgNPs) have garnered significant attention because of their distinctive biological and physicochemical properties. AgNPs are useful in various applications, such as biomedicine, water treatment, catalysis, biosensing, and electronic and magnetic devices [15]. Many studies have demonstrated that the utilization of Ag inhibits the activities of pathogenic microorganisms, such as bacteria, viruses, fungi, insects, and nematodes [16]. AgNPs mainly target pathogenic microorganisms by affecting the cell wall (enhancing its permeability and releasing cell wall components), mitochondria (disrupting ATP generation), protein (breaking disulfide or sulfhydryl bonds), and DNA (binding to sequences) [17]. In the past decade, AgNPs have been widely reported, particularly in the fields of health and medicine, as entirely new antibacterial agents [18].

Antimicrobial resistance is emerging as a global public health concern. Specifically, the global issue relates to antibiotic resistance and the proliferation of multidrug-resistant bacteria [19,20]. A recent report from the Centers for Disease Control and Prevention indicated that over 2.8 million antibiotic-resistant infections occur annually in the United States, resulting in more than 35,000 deaths [21]. Among the antibiotic-resistant pathogens, *Staphylococcus aureus* has become a leading pathogen worldwide, causing serious health disorders in humans [22]. Therefore, the development of novel and effective antibiotics against pathogenic bacteria in humans is a constant high-level need.

The present study addresses this gap by focusing on *Aggregatimonas sangjinii* F202Z8^T^, a Gram-negative bacterium isolated from a rusty iron plate from the intertidal region of Taean, the Republic of Korea [23]. The choice of *A. sangjinii* F202Z8^T^ was based on its origin in a metal-rich environment, suggesting its potential for metal NP synthesis. Our study aims to explore and harness this potential, positioning it at the forefront of innovative approaches in the field of green NP synthesis. The synthesized AgNPs were characterized in terms of their antimicrobial efficacy against pathogenic bacteria, an area of significant global concern due to the rise in antibiotic resistance. This study not only contributes to the green synthesis of AgNPs but also addresses the urgent need for new antimicrobial agents, marking a significant advancement in both the fields of NP synthesis and public health for application in medical science.

## 2. Materials and Methods

### 2.1. Bacterial Strains

*A. sangjinii* F202Z8^T^ was isolated from a rusty iron plate continuously exposed to seawater in the intertidal region of Taean, the Republic of Korea (36°35′25″ N, 126°17′17″ E) on 1 June 2018 [23]. The *A. sangjinii* F202Z8^T^ strain was routinely cultured on Marine Broth 2216 (Difco, Franklin Lakes, NJ, USA) with 1.5% agar at 27 °C. The strains used for antibacterial assays were *Bacillus subtilis* Korean Collection for Type Culture (KCTC) 3135^T^, *Escherichia coli* KCTC 2441^T^, *Salmonella typhimurium* KCTC 1925^T^, and *Staphylococcus aureus* American Type Culture Collection (ATCC) 6538^T^. Prior to the experiments, fresh cultures of each bacterial strain were prepared using Luria–Bertani (LB) medium with 1.5% agar at 30 °C (37 °C for *E. coli*).

### 2.2. Synthesis of Silver Nanoparticles

The synthesis of AgNPs using the strain F202Z8^T^ was examined using a slight modification of the method from Srivastava et al. [24]. In brief, the strain F202Z8^T^ was cultured in 200 mL of marine broth at 27 °C for 3 d, and then the cells were harvested through centrifugation at 10,000× *g* for 10 min. The pellets and supernatants were used for intracellular and extracellular synthesis, respectively. The pellets were thoroughly washed thrice with phosphate-buffered saline (137 mM NaCl, 2.7 mM KCl, 4.3 mM Na_2_HPO_4_, and 1.47 mM KH_2_PO_4_; pH 7.4) and twice with sterilized fresh water to remove any media components from the cells, and 0.4 g of wet weight pellets was mixed separately with an equal culture volume of 1 mM AgNO_3_ solution. The supernatants were mixed separately with equal volumes of 1 mM AgNO_3_ solution. All the reaction mixtures were incubated at 27 °C for 36 h at 150 rpm. Two controls, one with no inoculum and one with heat-killed cells as the inoculum, were incubated under the same conditions. After 36 h of incubation, the reaction mixtures turned dark brown, indicating AgNP formation. For intracellular NPs, the pellets and the supernatants were retrieved via centrifugation at 5000× *g* for 10 min at 20 °C; the collected pellets were extracted using the solvent or ultrasonication methods as described previously [24,25]. The dark brown solution obtained from the intracellular methods was centrifuged at 20,000 rpm for 15 min at 20 °C, and the precipitate was washed twice with sterilized fresh water to remove any media components. They were lyophilized to obtain the powder form and were further characterized and tested for biological activity.

### 2.3. Characterization of Silver Nanoparticles

UV–vis spectral analysis of the biosynthesized AgNPs was performed using a spectrophotometer (SpectriMas i3x, Molecular Devices, Sunnyvale, CA, USA), and the absorbance was measured at wavelengths in the range of 300–700 nm.

The size and concentration of the biosynthesized AgNPs were assessed using nanoparticle tracking analysis (NTA; Malvern Panalytical, Malvern, UK). NTA was performed using a NanoSight LM10 instrument equipped with a 405 nm violet laser and a CMOS camera (Hamamatsu Photonics, Shizuoka, Japan). The AgNPs were diluted 1000-fold with 1× sterilized PBS buffer filtered through a 0.2 μm sieve; the samples were further loaded into a flow-cell top plate using a syringe pump. All videos of 30–60 s duration at 25 °C were recorded and processed using NTA software (version 3.1). All measurements were repeated thrice.

The morphology of the biosynthesized AgNPs was examined using transmission electron microscopy (TEM; JEM-2100, JEOL Ltd., Tokyo, Japan) at an acceleration voltage of 100 kV. A drop of AgNPs was loaded onto carbon-coated Formvar grids (EMS, Hatfield, PA, USA) and air-dried for 10 min before measurements. The structural morphology of the biosynthesized AgNPs was generated using a Cs-TEM (JEM-ARM200F, JEOL Ltd., Tokyo, Japan) equipped with an energy-dispersive X-ray spectroscopy (EDS) attachment at an acceleration voltage of 200 kV. The EDS analysis was conducted at 10 different points, and the results of these elements are expressed as average values.

The functional groups of the synthesized AgNPs were determined using Fourier transform infrared (FTIR) spectrometry (PerkinElmer Inc., Waltham, MA, USA). The spectrum was recorded in the range of 400–4000 cm^−1^ at a resolution of 4 cm^−1^ to analyze the surface capping of the AgNPs. The spectra obtained were plotted as transmittance (%) versus wavenumber (cm^−1^).

### 2.4. Antibacterial Tests of Gold and Silver Nanoparticles

The antibacterial activity of the biosynthesized AgNPs was determined using a disc diffusion test [26]. To examine the antibacterial effect of AgNPs on four pathogenic bacteria, approximately 5 × 10^5^ colony-forming units (CFU) of the pathogenic bacteria were spread on LB agar plates, and then paper discs with a diameter of 8 mm containing 5, 10, and 20 µg/mL of AgNPs were gently placed on agar plates. The concentration of the synthesized AgNPs was determined using a high-accuracy weighing balance (Cubis II, Sartorius AG, Gottingen, Germany), ensuring accurate measurement of total AgNP weight. A paper disc containing the nutrient medium or 1 mM AgNO_3_ solution was used as the control. The culture plates were incubated at 30 °C (37 °C for *E. coli*) for 24 h.

To examine the minimum inhibitory concentration (MIC) and minimum bactericidal concentration (MBC) [27], the growth rates of the pathogenic bacteria exposed to various concentrations of AgNPs were measured at 600 nm using a personal bioreactor (RTS-1, BioSan, Riga, Latvia) at 30 °C (37 °C for *E. coli*) for 36 h. The MIC and MBC values were defined as the lowest concentration of the sample that showed no growth and a concentration where growth was not observed up to 36 h, respectively. All antibacterial test experiments were performed in triplicate.

## 3. Results and Discussion

### 3.1. Biosynthesis of AgNPs

The biosynthesis of AgNPs using both intracellular (bacterial cells) and extracellular (supernatants) methods was performed separately by observing the color changes of the mixed samples after the addition of AgNO_3_ solution from a dark brown color. The color changed because of the surface plasmon resonance (SPR) of the biosynthesized AgNPs [28]. After 36 h of reaction in the intracellular sample, the initially uncolored solutions turned dark brown, indicating the formation of AgNPs (Figure 1). The mixture used for the extracellular synthesis of AgNPs did not exhibit any color change. The controls containing heat-killed bacterial cells or supernatants with AgNO_3_ solution also did not show any color change. There have been a few studies elucidating the mechanism of AgNP synthesis by bacteria. Bacterial AgNPs are synthesized through the bio-reduction of Ag^+^ ions, which are bio-catalyzed by NADPH and NADPH-dependent enzymes such as nitrate reductase [29]. In addition, the formation of AgNPs on the cell surface might be attributed to the electrostatic interactions between the Ag^+^ and the negatively charged carboxylate groups present either on specific enzymes, proteins, pigments, or even polypeptides of the cell wall [30]. However, the mechanism of biological AgNP synthesis remains largely unknown. Previously, we identified 17 putative genes involved in the AgNP synthesis based on the gene annotation analyses in the complete genomic sequence of the strain *A. sangjinii* F202Z8^T^ [23]. These genes have the potential functions of stimulating reductive factors (alkaline phosphatase, etc.), stimulating the reduction of Ag^+^ to Ag^0^ (nitrate reductase), constituting the components of the electron transport chain in the respiratory pathway (C-type cytochrome, etc.), indirectly maintaining reduction conditions, and regulating enzyme activity (glutathione synthase, etc.) [31,32]. Although the role of each gene has not been fully characterized, these genetic data might serve as clues to the mechanism of bacterial AgNP synthesis.

### 3.2. Characterization of AgNPs

Typical metallic AgNPs exhibit maximum absorption wavelengths in the range of 400 to 450 nm [33]. In our study, the AgNP spectrum showed two broad bands in the visible range at 390 nm and 465 nm (Figure 1). These peaks could result from the capping properties of AgNPs, which are known to display SPR at the specified wavelengths, or they could be a result of our synthesized NPs forming a core–shell structure with different biomolecules like proteins [34,35]. The absorption peak at 465 nm was attributed to the heterogeneous size distribution of the AgNPs [36]. This observation indicates that the AgNPs synthesized using *A*. *sangjinii* F202Z8^T^ occurred intracellularly.

The size, morphology, and distribution of the biosynthesized AgNPs were determined using NTA, TEM, and EDS analyses. The AgNPs were spherical and well dispersed without notable agglomeration (Figure 2A) and were 27–82 nm in size, with an average diameter of 49 nm (Figure 2B). Metallic AgNPs typically exhibit a characteristic optical absorption peak at approximately 3 keV due to SPR [37]. Elemental mapping revealed that the most abundant element in the AgNPs was silver, at approximately 3 keV (Figure 2C), similar to the previous reports [30,38,39]. The wt% of Ag, carbon (C), and oxygen (O) elements was 77.51%, 14.83%, and 7.66%, respectively.

The FTIR spectrum showed the presence of functional groups in the biosynthesized AgNPs, which were measured using absorbance bands from 400 to 4000 cm^−1^. FTIR analysis was used to determine the connections between Ag and bioactive materials responsible for the creation and stability of NPs as capping agents [40]. The spectrum pattern showed different peaks at 3210, 2959, 2050, 1623, 1517, 1339, 1077, 834, 531, 443, and 414 cm^−1^ (Figure 3). The high absorption peak at 3210 cm^−1^ was attributed to the O–H hydroxyl group of the carboxylic acid, which in turn was attributed to the N–H group of the amides [41]. The absorption peak at 2959 cm^−1^ was attributed to C–H stretching vibration in the alkyl group, specifically in the methyl group (CH_3_) [42], and the peak at 2050 cm^−1^ was attributed to the C=C group of alkenes. The absorption peak at 1623 cm^−1^ was attributed to the C=O carbonyl group, while the peak at 1517 cm^−1^ was attributed to the C=N bond of the secondary amide. These groups indicate the presence of proteins capable of reducing and stabilizing the AgNPs. The absorption peak at 1339 cm^−1^ was attributed to CH_3_ group bending or the NO_2_ group [43]. The absorption peak at 1077 cm^−1^ was attributed to C–O–C stretching. The absorption peak at 834 cm^−1^ was attributed to the bending vibration of Ag–O, which may contribute to the stabilization of the AgNPs, and the peak at 531 cm^−1^ was attributed to C–Br stretching. These proteins or enzymes are responsible for providing the translation and stabilization of microbial-mediated AgNPs [40]. In addition, these proteins or peptides can also bind to AgNPs via electrostatic attraction from the negatively charged carboxylate groups in the existing enzymes within bacterial cell walls [44]. The detection of functional groups associated with proteins indicated that these biomolecules could be involved in the synthesis of AgNPs and in capping the produced Ag as a stabilizing agent [30,40,45].

### 3.3. Antibacterial Activity of Biosynthesized AgNPs

The antimicrobial activity of the biosynthesized AgNPs was evaluated against four pathogenic bacteria, *E. coli*, *S. typhimurium*, *B. subtilis*, and *S. aureus,* using the disc diffusion test (Figure 4A). AgNPs showed a dose-dependent increase in clearing zones against all pathogenic bacteria. In particular, they showed the highest inhibition against *S. aureus*, which causes serious health disorders worldwide [22].

The relative rate and extent of the bactericidal activity of AgNPs were evaluated by observing the dose-dependent growth kinetics of the pathogenic bacteria, except for *S. typhimurium*, which exhibited the lowest activity. The growth profiles of the pathogenic bacteria treated with different concentrations of AgNPs are shown in Figure 4B. The results showed that the growth kinetics of all pathogenic bacteria were influenced by the addition of AgNPs compared to the negative control grown without AgNPs. The growth of pathogenic bacteria decreased with increasing AgNP concentration; however, the inhibitory effect recovered over time, and the bacteria showed late growth. Although the recovery of pathogenic bacteria occurred with AgNPs, no visible growth of pathogenic bacteria was observed up to 36 h, which represents the bactericidal concentration at their respective MBC values [46]. Based on the results of the growth kinetics, the biosynthesized AgNPs exhibited MICs of 80, 40, and 30 µg/mL for *E. coli*, *B. subtilis*, and *S. aureus*, respectively, which are consistent with the results of the disc diffusion test. These results indicated that biogenic AgNPs can inhibit the growth of these three pathogenic bacteria. Notably, the degrees of inhibitory effects of AgNPs on bacterial growth in our study were significantly higher than those of other known antibacterial agents [27,47,48]. The MBC values of biosynthesized AgNPs were 100, 50, and 40 µg/mL for *E. coli*, *B. subtilis*, and *S. aureus*, respectively, which are lower than those observed for *E. coli* and *S. aureus* in a previous study [27]. One of the possible explanations for the lower MICs and MBCs of AgNPs relative to the known antibacterial agents is that the AgNPs may act as vehicles for the more efficient delivery of Ag^+^ to the bacterial membrane and cytoplasm [49]. The key factor for the antibacterial effect of AgNPs within the 20–80 nm range is attributed to the infiltration of bacterial cell envelopes by Ag^+^ ions released from the AgNPs [48]. Therefore, the size range of the F202Z8^T^-formed AgNPs (27–82 nm) might also be related to the higher antibacterial effect of AgNPs. Generally, the antibacterial effect of AgNPs on Gram-negative bacteria is stronger than that on Gram-positive bacteria due to the difference in the thickness of the cell membrane [50]. The cell membrane of Gram-positive bacteria, mainly composed of peptidoglycan, is approximately 30 nm thick, whereas that of Gram-negative bacteria is approximately 3–4 nm thick [51]. In contrast, the F202Z8^T^-formed AgNPs showed a stronger antibacterial effect on Gram-positive bacteria (*B. subtilis* and *S. aureus*) than on Gram-negative bacteria (*E. coli* and *S. typhimurium*). We speculate that the unique features of *S. typhimurium*, including the outer membrane structure or other defense mechanisms, might be responsible for the reduced susceptibility to AgNPs [52]. Additional studies are needed to investigate the antibacterial activity of AgNPs, focusing on the interaction mechanisms with different bacterial species, including *S. typhimurium*, to provide a more comprehensive understanding. *A. sangjinii* F202Z8^T^, used in this study, belongs to the family *Falavobacteriaceae*, which is the largest family in the phylum Bacteroidota (synonym Bacteriodetes). The family contains at least 153 genera and more than 700 species. Members of the family are found in a wide variety of marine, freshwater, and soil ecosystems, and some are also associated with animals or plants. Despite this diversity, the biosynthesis of AgNPs within these members has not been reported so far.

## 4. Conclusions

This paper presents a novel approach for the biosynthesis of AgNPs using *A. sangjinii* F202Z8^T^, a marine bacterium isolated from a unique environment. This marks the first utilization of this bacterium for NP synthesis, emphasizing our innovative approach to leveraging unique marine microorganisms for nanotechnological applications. The AgNPs were characterized using their specific size range (27–82 nm) and spherical morphology, showcasing distinct physical properties that were a direct result of the intracellular biosynthesis process, free from external stressors or additives. The functional characteristics of these AgNPs, particularly their broad-spectrum antibacterial activity against pathogenic Gram-positive and Gram-negative bacteria, represent a significant advancement in the field. The efficacy of using these AgNPs against *E. coli*, *B. subtilis*, and *S. aureus*, with particular emphasis on their minimum inhibitory concentrations and bactericidal capabilities, positions them as potential novel agents in combating antibiotic-resistant pathogens, addressing a critical global health concern. Furthermore, the synthesis method employed here, which is environmentally benign and cost-effective, aligns with the ongoing shift toward green nanotechnology. The approach demonstrates the potential of biological methods over traditional physical and chemical techniques, which are often resource-intensive and environmentally detrimental. 

In conclusion, our findings introduce *A. sangjinii* F202Z8^T^ as a novel candidate for efficient AgNP synthesis, with significant implications for the development of new antimicrobial strategies. The unique characteristics of the AgNPs synthesized in this study, coupled with their demonstrated antibacterial efficacy, underline the potential of this approach in both advancing nanoparticle research and addressing urgent public health needs. Future studies should focus on further exploring the capabilities of marine microorganisms in nanoparticle biosynthesis and expanding the applications of biogenic AgNPs in various fields beyond medicine, such as biotechnology, environmental remediation, and nanodevice fabrication.

## Figures and Tables

**Figure 1 microorganisms-11-02975-f001:**
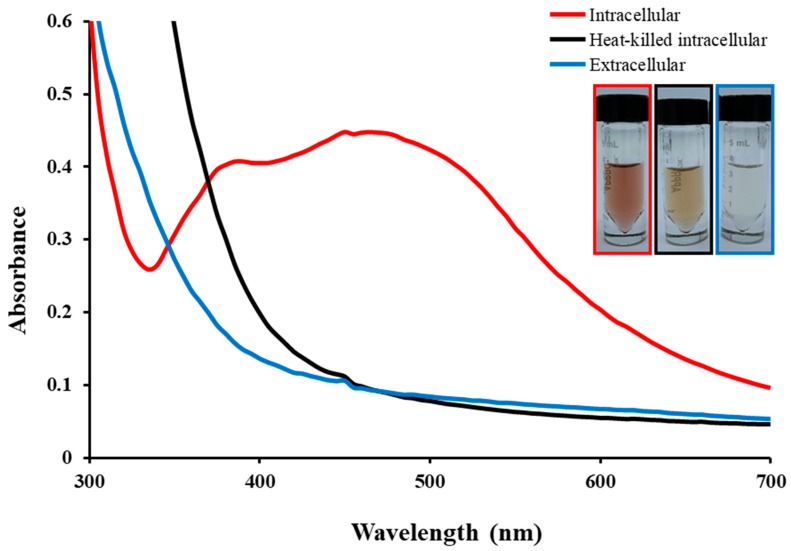
UV–visible absorbance spectra of biogenic AgNPs synthesized intracellularly using *Aggregatimonas sangjinii* F202Z8^T^ (intracellular, the red line) pellet with AgNO_3_; (intracellular, the black line) heat-killed pellet with AgNO_3_ as control; (extracellular, the blue line) supernatant with AgNO_3_.

**Figure 2 microorganisms-11-02975-f002:**
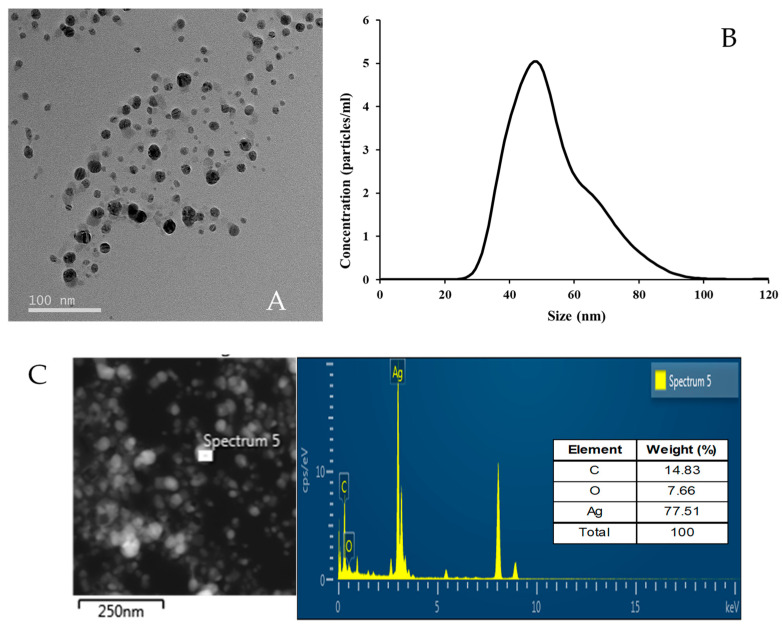
Morphology, size, and distributed element of the AgNPs biosynthesized using *A*. *sangjinii* F202Z8^T^. (**A**) Transmission electron micrograph image (scale bar, 100 nm) of the AgNPs; (**B**) size distribution of the AgNPs based on nanoparticle tracking analysis; (**C**) EDS spectrum of the AgNPs.

**Figure 3 microorganisms-11-02975-f003:**
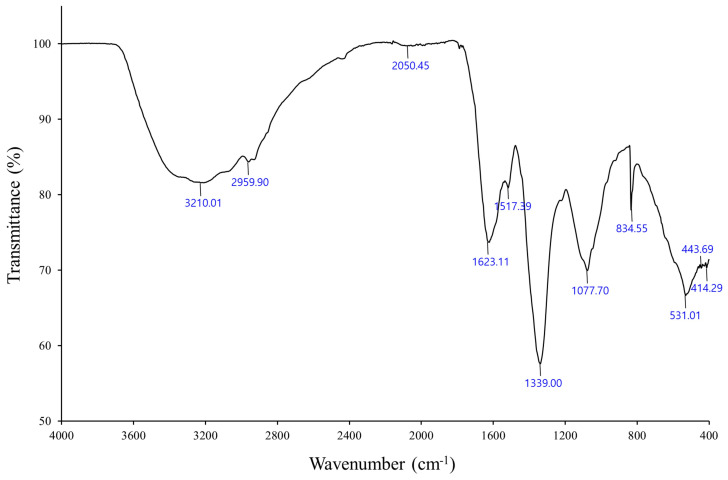
Fourier transform infrared spectrum showing functional groups responsible for the synthesis and stabilization of the AgNPs biosynthesized using *A*. *sangjinii* F202Z8^T^.

**Figure 4 microorganisms-11-02975-f004:**
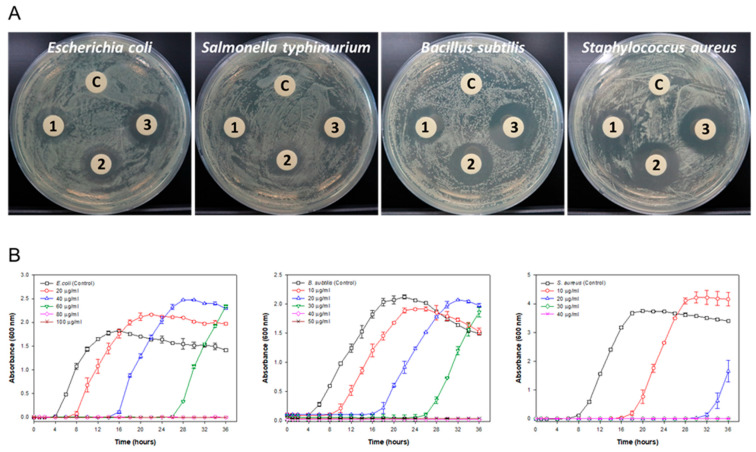
Antibacterial activities against pathogenic bacteria in the presence of varying concentrations of the AgNPs biosynthesized using *A*. *sangjinii* F202Z8^T^. (**A**) Disc diffusion test (from left to right: *Escherichia coli* KCTC 2441^T^, *Salmonella typhimurium* KCTC 1925^T^, *Bacillus subtilis* KCTC 3135^T^, and *Staphylococcus aureus* ATCC 6538^T^). C, Control (medium containing 1 mM AgNO_3_); 1, AgNPs 5 µg/mL; 2, AgNPs 10 µg/mL; 3, AgNPs 20 µg/mL. (**B**) Growth profiles against pathogenic bacteria (from left to right: *E. coli* KCTC 2441^T^, *B. subtilis* KCTC 3135^T^, and *S. aureus* ATCC 6538^T^) in the presence of varying concentrations of AgNPs.

## Data Availability

The data presented in this study are available on reasonable request from the corresponding author.

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
