# Peer review of "Synthesis of Silver Nanoparticles Using Aggregatimonas sangjinii F202Z8T and Their Biological Characterization"

_microorganisms, 2023, doi:10.3390/microorganisms11122975_

Round 1

Reviewer 1 Report

Comments and Suggestions for Authors

The manuscript entitled “Synthesis of silver nanoparticles using Aggregatimonas sangjinii F202Z8T and their biological characterizationis aims to intracellularly AgNPs synthesis using strain Aggregatimonas sangjinii F202Z8T and its the antibacterial activity investigation against various pathogenic Gram-positive and Gram-negative bacteria ( Escherichia coli, Bacillus subtilis, and Staphylococcus aureus). To my mind this manuscript is topical and corresponding to the aims and scopes of the Microorganisms journal.

Here are the comments I found while reading the manuscript

1. The abstract should more specifically describe the results of the synthesis of AgNPs using strain Aggregatimonas sangjinii F202Z8T

2. The introduction should clearly describe the purpose of the work and its novelty. A huge amount of work on the green synthesis of AgNPs has been carried out to date. It is necessary to note the novelty of the authors’ work. And on the basis of novelty, I would advise supplementing the introduction by bringing it to the work of the authors. So far it seems too general.

3. Since during the synthesis of AgNPs in media containing chlorine and phosphate ions there is a problem of silver precipitation in chemical precipitates, it is necessary to describe in detail the specific composition in which the bacterial synthesis of AgNPs took place. Often authors use chloride-free and phosphate-free solutions. The same goes for assessing antibacterial activity.

4. It is worth checking the style of the language. For example 113 “freshly prepared”

5. disc diffusion test, in my opinion, does not provide adequate data on poorly soluble antimicrobial agents. Ionic silver can interact with cellulose agar, especially if it is stabilized by a jacket of polysaccharides or proteins during bacterial synthesis. It is necessary to conduct experiments with cultures in solution.

6. 164-165 – there are a significant number of synthesis mechanisms. Have the presence and activity of nitrate reductase been determined for your strain?

7. Was the concentration of silver in the solution measured after contact with the cells?

8. The FTIR spectrum description does not provide new data to describe stabilization in bacterial synthesis. It is clear that there is a lot of organic matter. Further understanding depends on whether they could be cleared of biomass and molecules that do not play a role in stabilization. Perhaps MALDI analysis could somehow answer these questions.

9. 211 alkane? Hydrocarbon chain.

10. Microbial particles often have a mono-size distribution. It is worth dwelling on this in detail. According to electron microscopy, there are very large particles that may have fallen out as a result of contact of silver with phosphates and chlorides from solution or cytoplasm. In general, I would like to see microtomized native cell preparations in SEM and PEM images

11. 246 80, 40, and 30 μg/ml particles added, given in terms of silver or particle weight

12. the conclusions must be reconsidered in terms of the novelty of the data obtained by the authors. I don't see anything new yet.

Comments on the Quality of English Language

Moderate editing of English language (style) required

Author Response

Dear Reviewer,

I sincerely appreciate your valuable feedback and insightful suggestions on my manuscript. Following your recommendations, I have thoroughly revised the document. Each of your specific comments has been addressed in detail, as outlined in the attached response file, where I have explained the changes made and provided to the revised sections. I have also made additional adjustments to enhance the clarity and coherence of the manuscript. I believe these revisions have significantly strengthened the manuscript and hope that it now meets the publication standards. Thank you for your guidance and consideration.

Reviewer 2 Report

Comments and Suggestions for Authors

Dear

The authors report the synthesis of silver nanoparticles using A. sangjinii F202Z8T, the research work is well presented, below I show my observations that I consider can help a better presentation of the research work.

Line 64.- It is said that studies have shown that silver nanoparticles inhibit the pathogenic activity of: bacteria, viruses, fungi, insects and nematodes. Place references to published works where the inhibitory effect is discussed, especially in insects and nematodes.

Line 214: It says amide II, do you mean secondary amide? I suggest putting "secondary amide" instead of amide II or you can even name imides.

Figure 2 A. I suggest replacing the TEM image with one of better quality and that provides more information, an HRTEM image would be very appropriate.

Figure 2 C. Only an EDS spectrum is taken in a single region, I suggest performing EDS at 10 points and obtaining the average of these EDS results, it is certain that the value of 91.11% of silver will change and a percentage of silver will be obtained more representative.

I consider it very important to provide an explanation as to why the inhibitory halo is not noticeable in Salmonella typhimurium. If it is mentioned that an important factor is the thickness of the cell wall in Gram (+) and Gram (-), an explanation should be sought. this microorganism.

Reference 24 does not correspond to what was described, this refers to Pseudomonas aeruginosa SM1 and you say that it is the previous report regarding A. sangjinii F202Z8T.

BEST REGARDS

Author Response

(The authors gave the same response as above.)

Round 2

Reviewer 1 Report

Comments and Suggestions for Authors

The authors significantly improved the manuscript in accordance with my recommendations. However, there are a few comments. First of all, it’s worthwhile to clearly write down the purpose of the work, since now it looks like an abstract. in the discussion of the results, more attention should be paid to comparing the obtained particles with particles obtained in other bacteria, since in my opinion, apart from the new strain isolated from technogenically polluted conditions, there is no novelty in this study. Authors should think more carefully about the novelty of their research and state it before the purpose and in the conclusion. In addition, it is worth clarifying whether it is a marine bacterium or not, the authors are talking about marine bacteria, and the strain was isolated from a rusty iron plate. It is worthwhile to describe in detail the location of isolation, including habitat conditions.

Comments on the Quality of English Language

 Minor editing of English language required

Author Response

The authors significantly improved the manuscript in accordance with my recommendations. However, there are a few comments. First of all, it’s worthwhile to clearly write down the purpose of the work, since now it looks like an abstract. in the discussion of the results, more attention should be paid to comparing the obtained particles with particles obtained in other bacteria, since in my opinion, apart from the new strain isolated from technogenically polluted conditions, there is no novelty in this study. Authors should think more carefully about the novelty of their research and state it before the purpose and in the conclusion. In addition, it is worth clarifying whether it is a marine bacterium or not, the authors are talking about marine bacteria, and the strain was isolated from a rusty iron plate. It is worthwhile to describe in detail the location of isolation, including habitat conditions.

: Thank you so much for your comments on our manuscript. We appreciate the enhancement of our manuscript following your recommendations.

As newly added in Abstract (Line 13-15), the aim of this study is to describe the general features and eco-friendly biosynthesis of AgNPs from the marine bacterium, Aggregatimonas sangjinii F202Z8T. To our knowledge, the biosynthesis of AgNPs using this novel strain has not been previously reported.

There are a few studies to describe antimicrobial activity of microbial AgNPs (Lines 262-274). However, we think the novelty of our study is supported by the novelty of this strain itself (a novel genus and the halophilic growth of this strain) and by description of the antimicrobial activity of AgNPs against various bacteria including pathogenic ones. We believe that the purpose and novelty of this manuscript are sufficiently expressed to suggest the potential of this strain to produce eco-friendly AgNPs.

Regarding the origin of this strain, the iron plate where we isolated this strain from was continuously exposed to seawater. Thus, we think this strain can be described as a marine bacterium. In addition, the halophilic growth of this strain (the optimum salinity was 4%) might be closely related to the ecological origin of this strain. We included the information of isolation location and the habitat conditions in Lines 96-97.

This manuscript underwent English language editing by a professional service prior to its initial submission, and proof of this is attached.
